# Identification of Single Yeast Budding Using Impedance Cytometry with a Narrow Electrode Span

**DOI:** 10.3390/s22207743

**Published:** 2022-10-12

**Authors:** Xun Liu, Tao Tang, Po-Wei Yi, Yapeng Yuan, Cheng Lei, Ming Li, Yo Tanaka, Yoichiroh Hosokawa, Yaxiaer Yalikun

**Affiliations:** 1Division of Materials Science, Nara Institute of Science and Technology, 8916-5 Takayama-cho, Ikoma 630-0192, Nara, Japan; 2Department of Applied Chemistry, National Yang Ming Chiao Tung University, Hsinchu 30010, Taiwan; 3Center for Biosystems Dynamics Research (BDR), RIKEN, 1-3 Yamadaoka, Suita 565-0871, Osaka, Japan; 4The Institute of Technological Sciences, Wuhan University, Wuhan 430072, China; 5School of Engineering, Macquarie University, Sydney 2109, Australia

**Keywords:** impedance cytometry, electrode gap, detection sensitivity, yeast cells

## Abstract

Impedance cytometry is wildly used in single-cell detection, and its sensitivity is essential for determining the status of single cells. In this work, we focus on the effect of electrode gap on detection sensitivity. Through comparing the electrode span of 1 µm and 5 µm, our work shows that narrowing the electrode span could greatly improve detection sensitivity. The mechanism underlying the sensitivity improvement was analyzed via numerical simulation. The small electrode gap (1 µm) allows the electric field to concentrate near the detection area, resulting in a high sensitivity for tiny particles. This finding is also verified with the mixture suspension of 1 µm and 3 µm polystyrene beads. As a result, the electrodes with 1 µm gap can detect more 1 µm beads in the suspension than electrodes with 5 µm gap. Additionally, for single yeast cells analysis, it is found that impedance cytometry with 1 µm electrodes gap can easily distinguish budding yeast cells, which cannot be realized by the impedance cytometry with 5 µm electrodes gap. All experimental results support that narrowing the electrode gap is necessary for tiny particle detection, which is an important step in the development of submicron and nanoscale impedance cytometry.

## 1. Introduction

Impedance cytometry is a powerful technology which is widely used in bio-detection, cells analysis [1], food security [2], and drug resistance [3,4] for advantages like precision, efficiency, and no mark required. With a low detection frequency (<1 MHz), it can easily measure the size [5], amount [6], shape [7], phenotypes [8], and position [9] of cells. By applying high-frequency detection voltage (>1 MHz), cell membranes could be partly conductive [10], and some intracellular components [10,11] and property [12] could be detected.

In terms of single cell detection, the sensitivity of impedance cytometry is quite important, as it determines the applicable size of single cells/particles [13]. To date, the sensitivity [14,15] has been shown to be related to channel dimension, electrode layout, fluid conductivity, and electrode dimensions. Any tiny changes in electrodes or microchannels would be shown on the magnitude of impedance pulses. Based on this phenomenon, a conventional method of improving the detection sensitivity is to widen electrodes or narrow detection channels for a strong electric field [15] as a result of sacrificing signal-to-noise ratio (SNR) [16]. In this work, we provided an alternative way to improve the detection sensitivity: through narrowing the electrode span. The electrode span was found to impact the local strength of electric filed in the detection area, which could facilitate the detection of the tiny objects in a large channel.

Typically, the electrodes in impedance cytometry are fabricated by photolithography and lift-off. However, the fabrication of small electrodes or gaps is difficult, and the defects easily occur in developing process in lift-off with a feature size of at least 1 µm. To avoid possible defects in photolithography, in this work, a femtosecond laser was applied to ablate the electrodes to generate the small gap. As femtosecond laser is widely used in micro-nano fabrication [17] and even cell perforation [18]. With high precious and low extra damage to materials, it is a potential option for electrode fabrication.

To verify the sensitivity of electrodes with a small gap, some indicator with a small difference in size is needed, such as yeast cells. Here, we chose the yeast *Saccharomyces cerevisiae*, which is widely used in the food industry and biology research [19]. In the cell cycle of *S. cerevisiae* yeast, single cells always turn into a budding situation with morphological changes [20,21], and the small difference in cell shape and size makes *S. cerevisiae* yeast become the well indicator for various cell sensors [8,22], and in our case. 

In this paper, we improve the sensitivity of impedance cytometry by decreasing the gap of electrodes to smaller than 1 µm, while electrodes have a width of 30 µm. To investigate the effects of gaps on the detection sensitivity, numerical models were built by finite elements methods to express how the smaller gap could enlarge the local strength of electric field to generate higher differential signals when objects pass through electrodes. To verify our theory, experiments were performed on small particles (1 µm and 3 µm beads) and single cells. As a reference, electrodes with 5 µm gaps were also employed in experiments. The gap of 1 µm was fabricated by femtosecond laser ablation on whole electrodes, while a 5 µm gap was directly fabricated by normal lift-off technology. In detection of polystyrene particles suspension with diameters at 1 µm and 3 µm, many more 1 µm particles could be detected and distinguished using electrodes with 1 µm gaps. Then, in the detection of yeast cells, the budding yeast cells could be successfully identified by the electrodes with 1 µm gap.

## 2. Materials and Methods

### 2.1. Microfluidic Chip

The microfluidic chip in impedance cytometry is assembled with PDMS structure and glass with coplanar electrodes by irreversible bonding after plasma treatment. The microchannel is obtained from casting mixed PDMS (SYLGARD 184, Dow Corning, Midland, MI, USA) on a SU8 (MicroChem Corp., Newton, MA, USA) mold by common soft lithograph. The electrodes, which consist of two layers, about 70 nm thick chromium (Cr) on glass and 70 nm thick gold (Au) on chromium, are all fabricated by photolithograph and lift-off. 

The microchannel in detection area is 12 µm wide and 10 µm high, while three coplanar parallel electrodes are 30 µm wide. In our research, there are two kinds of gaps of designed electrodes—1 µm and 5 µm. The 5 µm wide gaps between electrodes are designed on mask initially and generated by photolithography, while 1 µm gaps are initially fabricated as one 92 µm wide electrode and then ablate two 1 µm wide gaps by femtosecond laser to separate the one electrode into three independent electrodes.

### 2.2. Sample Preparation

In beads experiments, the mixed beads suspension contained polystyrene particles with diameters of 1 µm and 3 µm. The ratio of 1 µm and 3 µm polystyrene beads in the suspension was around 9:1.

In single cell detection, *S. cerevisiae* yeast cells were employed, and all cells were pre-mixed with polystyrene particles in a diameter of 3 µm before experiments. The beads mixed in cells are used as reference. In detail, yeast cells were firstly diluted in 1× PBS solution and then put in a thermostat at 38 °C for 3 h awake. Figure 1 shows the photos of particles and yeast cells under a microscope, and normal yeast cells have a larger size than 3 um particles, while the budding yeast cells are larger than normal yeast.

Before experiments, all samples were transferred to a 1× PBS solution with 0.1% Tween 20 solution to avoid sample aggregation. Sample suspension was loaded into the microchannel using a syringe pump with a flow rate of 10 µL/min. 

### 2.3. Experiments Setup and Data Process

In detection, a 3.3 V voltage signal with two frequencies (500 kHz and 600 MHz) was generated by a FPGA-based lock-in amplifier (Diligent Eclypse Z7, Houston, TX, USA) and connected the central electrode. Current signals propagated from the central electrode to both detection electrodes through the conductive fluids in the microchannel. When single objects flow through the detection area, the induced current fluctuation on two detection electrodes was detected and converted to voltage signals by a I/V converter. A differential amplifier was employed here to compare the signal fluctuation on two electrodes and transmit the results to the lock-in amplifier. In the lock-in amplifier, current signals were deconvoluted to the impedance signals at two frequencies, which were then collected by a data collector (USB-6363 BNC, National Instruments, Austin, TX, USA) at a sampling rate of 125 kHz and displayed in NI DAQExpress (National Instruments, USA) on PC in the meantime.

To analyze the results, recorded data were imported to MATLAB first to characterize the impedance signals via two dielectric properties, i.e., the electrical diameter and tilt index. All signal processes were performed by our lab-made scripts. Besides, the electrical diameter [23] is defined as the cubic root of the impedance amplitude, which has been proved to be linearly correlated with the physical diameter of single objects. The tilt index [7,24] is defined as the ratio of left-half impedance pulse to the right-half and minus one, which is used to characterize the shape of single objects. By gathering these features of signals, distribution of features can be obtained to reveal the morphology and size of the objects.

## 3. Results and Discussion

### 3.1. Numerical Investigation of Impedance

To investigate the impact of the electrode span on the impedance detection and sensitivity, numerical models were built by commercial software, COMSOL Multiphysics 6.0 (COMSOL Inc., Burlington, MA, USA) and the AC module. In the 2D simulation model, two ground electrodes laid on two sides of positive electrodes with a gap of 1 µm or 5 µm, respectively. The electrode was set to be 30 μm. The area of channels (10 μm high) above electrodes was the detection area and set as 1× PBS. The relative permittivity and conductivity of 1× PBS in the microchannel were 80 and 1.4 S/m, respectively. The bead model was simplified as a 2D nonconductive circle. 

As a result, in the absence of objects in the detection area, the currents detected from two ground electrodes were equal, resulting in no impedance signals. As particles or cells passed the electrode span, the volume of conductive PBS solution above the span was replaced by the nonconductive object, inducing the sudden increase in the system impedance, i.e., impedance pulses. This is because the nonconductive objects interfere with the current propagation above the electrode span. It is worth noting that the majority of current propagation took place near the electrode span in the fluid. As shown in Figure 2a, a strong electric field was observed near electrode span. The differential current is the difference of the current from the two cathodes; due to the 2D geometry in simulation, the unit of current here is A/m. 

To date, according to several impedance-related studies [23,25], the 2D simulation is sufficient to analyze the relationships between impedance signals, single objects, and the detection electrodes. Therefore, in this work, we did not employ other methods (i.e., the analytical solution) to verify the simulation results. 

Figure 2a shows the 2D simulation of the impedance cytometry in our research, and the gaps of anode and each cathode were 5 µm (left) and 1 µm (right) here. The color corresponded to the voltage distribution that diminishes from the anode to the cathode. When a 3 µm particle moved through the detection area, the impedance magnitude changed according to the different heights of the particle in the channel (i.e., 3 µm, 5 µm, 7 µm to channel bottom). In other words, the impedance signals of single particles are trajectory-dependent. As particles get closer to the channel bottom, they induce higher impedance pulses. Due to the nonuniform strength of the electric field in microchannels, the current from anode propagates around the channel bottom. Additionally, when measuring a particle along the same trajectory, the highest impedance magnitude was measured for particles traveling through the 1 µm electrode span. 

Figure 2b shows the strength of electric field at different depths at the center of the electrode span. Compared to 5 µm gap, a 1 µm gap can result in a greater gradient of voltage between the cathode and anode, resulting in a higher strength of electric filed. In other words, there was a concentration of current near the 1 µm electrode span. With distance from the channel bottom, this field strength gradually dims. Interestingly, at a distance of 4 µm from the channel bottom, both types of electrode spans showed similar field strengths. This phenomenon indicates that regardless of the electrode span, both designs show the same sensitivity to single objects beyond 4 µm from the channel bottom, while the 1 µm electrode span provides a greater sensitivity within 4 µm.

Figure 2c,d illustrate the changes in the impedance magnitudes when measuring 1 µm and 3 µm particle, respectively. In Figure 2c, the impedance magnitudes of 1 µm particles varied between 0.02 and 0.28, in the case of the 1 µm electrode span. By contrast, this value varied between 0.02 and 0.09, in the case of the 5 µm gap. An impedance pulse that was three times higher was obtained by the 1 µm gap compared to the 5 µm gap. Considering background noise in real experiments, small impedance pulses are difficult to be detected. Therefore, we assume that using the 1 µm electrode gap, the impedance pulses of tiny particles (e.g., 1 µm) could easily be seen instead of being buried in background noise. Additionally, when measuring the 3 µm particle, the 1 µm electrode span can also provide the highest impedance magnitude in comparison to the 5 µm electrode span. Notably, the lowest impedance magnitude for both types of electrode spans were the same, which is because the field density (>4 µm from the channel bottom) of both designs was the same (see Figure 2b).

In this 2D simulation, the electric field strength can be increased by reducing the electrode gap, resulting in a stronger impedance pulse of single objects. Objects with strong impedance pulses could be easily detected by the system, for example, a high amplitude makes it easy to distinguish small objects from noise and to detect them accurately.

Notably, the 2D simulation results here only indicated that narrowing the electrodes would improve detection sensitivity. Optimal electrode spans for detecting particles with different sizes requires further research. Furthermore, the electrode size is also a key parameter for detection sensitivity that was found to increase with increasing electrode size [15]. In order to study the impact of electrode span, all electrodes were designed to be 30 µm wide. Besides, the electrode length in this work was limited by the channel width (12 µm) it is not a key parameter for detection sensitivity, which has been verified in other work [26]. Additionally, in this work, all electrodes were fabricated as traditional coplanar electrodes to eliminate the impact of electrode morphology [27] on the detection sensitivity. 

### 3.2. Single Beads Detection

To verify the impact of electrode span on the impedance detection, experiments were first performed on the mixture suspension of 1 µm and 3 µm beads. The impedance signals of both beads were calibrated to the electrical diameter, as shown in Figure 3a,b. Figure 3a indicates that the 5 µm gap was applicable for the detection of 1 µm and 3 µm beads, while there was an overlay area between both regarding the electrical diameter. Besides, using a 5 µm electrode gap was more likely to fail to detect some 1 µm beads than using 1 m electrode gap; for example, in Figure 3b, there were many more 1 µm beads that appeared in the distribution plot when using 1 µm electrode gap compared to a 5 µm gap. In this work, the ratio of 1 µm and 3 µm beads in the suspension was around 9:1. When using a 1 µm gap, the measured ration was close to the real value, but it did not work with a 5 m gap. This phenomenon was attributed to most impedance pulses of 1 µm beads being covered by the noise, as their magnitudes were too small.

Additionally, the electrical diameters of 1 µm and 3 µm beads distributed compactly and did not spread out; this result indicated that the effect of individual particles could be ignored in analysis. When the impedance values were impacted by the particle trajectory, their impedance metrics (i.e., electrical diameter) were distributed widely, and sometime, their electrical diameter was not matched with the real diameter of particles [9,23]. In this work, the microchannel was designed to be 12 µm wide and 10 µm high, which successfully limited the trajectory variation of single particles, resulting in the compact distribution of electrical diameters.

In Figure 3c,d, we employed the single-peak Gaussian model to discriminate the 1 µm beads from 3 µm beads based on their electrical diameter. Although it is possible to separate the two sizes of beads in both cases, there would be more overlapped area between two data sets when using 5 µm electrode span (see Figure 3c). The distribution of 1 µm beads was not perfectly isolated and spread out. By contrast, the separation of the two data sets with less overlap could be realized when using 1 µm electrode span for measurement (see Figure 3d). Interestingly, when using 1 µm electrode span, the electrical diameter (about 1.3 µm) of 1 µm beads was slightly larger than their physical diameter. This is possibly because some beads flowed near electrodes in detection, and their impedance magnitude was enhanced greatly. 

After the isolation, the ratio of 1 µm and 3 µm detected beads was about 4.52:1 (Real: 9:1), which means that half of 1 µm particles were detected using 1 µm electrode gap. By contrast, the ratio obtained by 5 µm span was about 0.14:1, which was far away from the real ratio (9:1). All beads’ experimental results could demonstrate that the electrodes with 1 µm gap can improve the detection sensitivity for small particles. 

### 3.3. Budding Yest Cells

The high sensitivity of impedance cytometry with 1 µm electrode span enables the detection of cells with small difference in size. In this work, yest cells were tested here, and 3 µm beads were employed as the reference for the calibration. Similar to most eukaryotic cells, yeast cells appear as morphological changes in the cell cycle [9,20], i.e., the cell budding. During the budding, the mother yeast cell produces a small bud that grows until it reaches a certain size and then separates [9]. 

In Figure 4a,b, impedance signals, collected from 5 µm and 1 µm gap, respectively, were sorted into two data groups following gaussian distribution; one group with smaller electrical diameters represents the 3 µm reference beads, whereas the other group represents yeast cells. Previous research [8] has shown that the budding cells would have bigger electrical diameters than normal yeast cell. Comparing Figure 4a,b, when the gap was set to 1 µm instead of 5 µm, the electrical diameter of 3 µm reference beads is much more centralized. Besides, in Figure 4b, when using 1 µm electrode gap, there was a small cluster of data outside of the yeast cells, indicating the existence of budding yeast cells. By contrast, when using a 5 µm electrode gap, the electrical diameter distribution of yeast cells was decentralized in a wide range from 5 µm to 8 µm in Figure 4a. There is no clear difference between normal and budding yeast cells in terms of yeast cells. In terms of detecting the same yeast cell suspension, the electrical diameter of yeast cells in Figure 4b could provide more details on the cell volume. This phenomenon can be attributed to the high sensitivity of impedance detection for the cell volume with 1 µm electrode span.

Figure 4c,d show the distribution of the tilt index of impedance pulses when measuring with 5 µm and 1 µm electrode gap. The tilt index was employed to characterize the shape of single objects. In theoretic, objects with symmetric shapes, such as spheres, should have a tilt index of around zero; objects with unsymmetric shapes would have a nonzero tilt index. The tilt index increases with increasing the unsymmetric level of single objects. Typically, yeast cells did not exhibit a perfect symmetrical shape as beads, thus, their impedance pulses would be asymmetric in shape, resulting in a tilt index away from the zero. In Figure 4e, the tilt index of thousands of yeast cells was not evenly distributed around zero, which was attributed to their asymmetric shape. In Figure 4f, we took advantage of the difference in cell size between yeast and budding cells to isolate the tilt index of budding cells from others; the tilt index of yeast cells and budding yeast cells ranged from about −0.2 to 0.4 and −0.1 to 0.3, respectively. There is no clear difference between budding or normal yeast cells regarding the tilt index. We think that the tilt index might not be sensitive enough to monitor the morphology change induced by the cell budding in this work. However, it is worth mentioning that the electrode span of 1 µm still performed better for the morphology determination, as the distribution of tilt index was more centralized when using 1 µm electrode span rather than 5 µm span. The photos of yeast sample in experiments can be found in Appendix A, and the number of budding yeast cells is lower than normal yeast.

Figure 5 shows the scatter plot of electrical diameter vs. tilt index. In Figure 5a, the impedance detection with 5 µm electrode span resulted in a large overlayed area between the reference beads and yeast cells. In contrast, Figure 5b is the scatter plot of impedance signals measuring from electrodes with a 1 µm gap. Three groups of points could be clearly distinguished, including 3 µm reference beads, yeast cells, and budding yeast cells, respectively. Here, a cluster of cells were found to have a bigger volume than other cells and were identified as the budding cells; this is because the budding cells typically had bigger cell volume, resulting in bigger electrical diameter in impedance detection. Besides, the microscopic images also indicated the existence of buddying cells in the detection suspension (see Appendix A). As a result, it is easy to conclude that the sensitivity of impedance cytometry could be improved through narrowing the electrode span. 

## 4. Conclusions

In summary, a small electrode gap in impedance cytometry can enhance the strength of electric field and increase the current density near the channel bottom. Target objects pass through the strengthened electrical field area, allowing the generation of larger impedance pulses than normal. This feature could facilitate the detection of small objects (e.g., ~1 µm diameter, such as bacteria or exosome) via the impedance detection with enhanced sensitivity. The only disadvantage of this method is the difficulty in fabrication, and in this work, the femtosecond laser was employed to fabricate the 1 µm electrode span.

The high sensitivity of impedance cytometry enables the discrimination of similar objects with small differences. For example, we employed the enhanced impedance cytometry to discriminate the budding yeast cells from normal ones in our experiments. This work provided a new path to improve the detection sensitivity of the impedance cytometry and prepare for the application in submicron and nano-scale detection. 

## Figures and Tables

**Figure 1 sensors-22-07743-f001:**
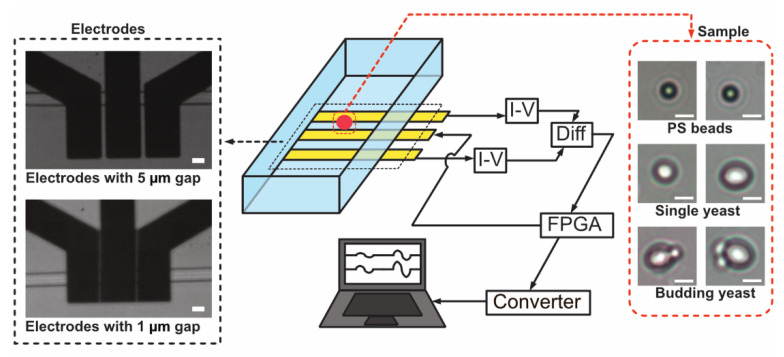
Demonstration of cell impedance detection in this research, sample conclude PS particles and yeast cells (scale bar: 5 µm); two kinds of electrodes have the gap of 5 µm and 1 µm (scale bar: 10 µm).

**Figure 2 sensors-22-07743-f002:**
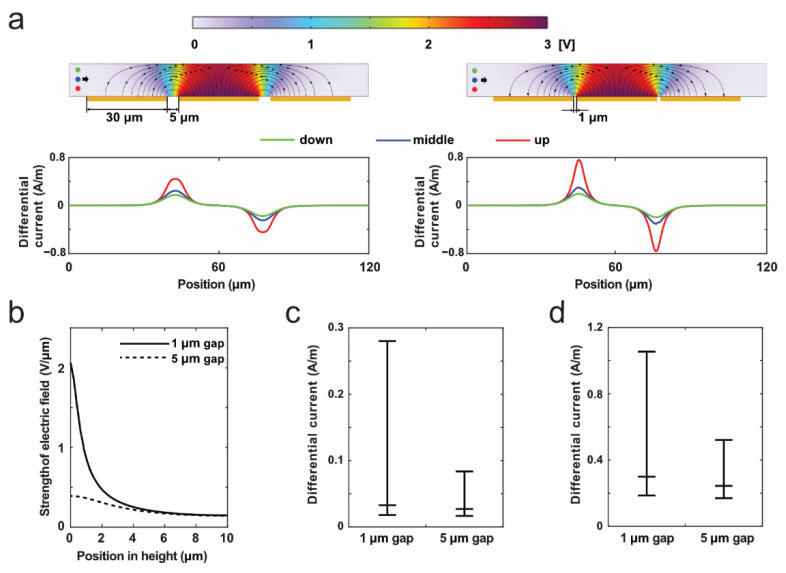
Simulation analysis of the impact of electrode span on the impedance detection. (**a**) Distribution of electric filed in the detection area of impedance cytometry (up), color represents the voltage, line with arrows means the current direction. The impedance pulses (down) were obtained when flowing 3 μm particle along three trajectories. The particles at different heights in the channel (i.e., 3 µm, 5 µm, 7 µm to channel bottom) were labelled in red, blue, and green, respectively. (**b**) Strength of electric field in the case of 1 µm gap and 5 µm gap. (**c**,**d**) Relationship between the impedance magnitude and the electrode gap, when measuring (**c**) 1 µm and (**d**) 3 µm particle. The impedance magnitude increases with decreasing the electrode gap.

**Figure 3 sensors-22-07743-f003:**
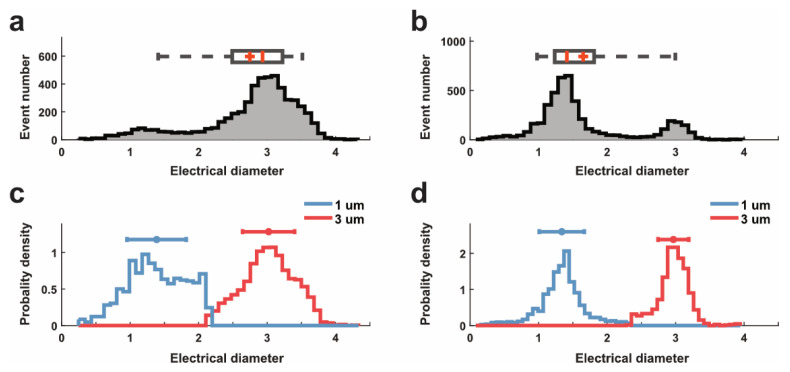
Electrical diameter of polystyrene beads when using different electrode spans for measurement. (**a**,**b**) Distribution of electrical diameter signals of detected beads in suspension when measuring with (**a**) 5 µm and (**b**) 1 µm electrode span. (**c**,**d**) The discrimination of two sizes of beads using the single-peak Gaussian model, based on their electrical diameters obtained with (**a**) 5 µm and (**b**) 1 µm electrode span. 1 µm beads were labelled in blue, and 3 µm beads were labelled in red.

**Figure 4 sensors-22-07743-f004:**
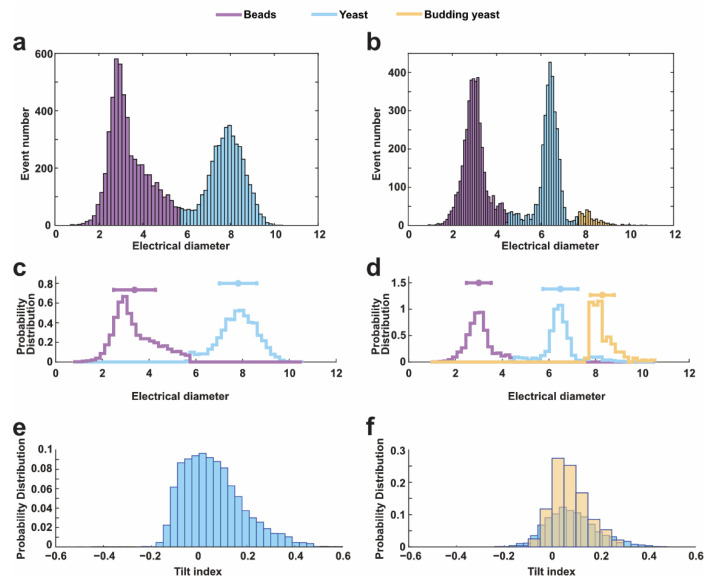
Dielectric properties of yeast cells when using different electrode spans for measurement, 3 µm beads were employed as the reference for calibration. (**a**,**b**) Distribution of electrical diameter signals of detected beads and cells in suspension when measuring with (**a**) 5 µm and (**b**) 1 µm electrode span. (**c**,**d**) The discrimination of reference beads and cells using the single-peak gaussian model, based on their electrical diameters obtained with (**a**) 5 µm and (**b**) 1 µm electrode span. (**e**,**f**) Probability distribution of the tilt index of impedance pulses obtained from electrodes with (**e**) 5 µm gap and (**f**) 1 µm gap.

**Figure 5 sensors-22-07743-f005:**
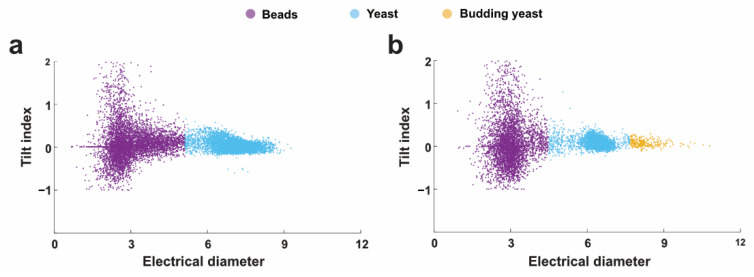
Scatter plot (electrical diameter vs. tilt index) of single cell detection; (**a**) signals from electrodes with 5 µm gap scatters; (**b**) signals from electrodes with 5 µm gap scatters.

## Data Availability

Not applicable.

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
