# Peer review of "Identification of Single Yeast Budding Using Impedance Cytometry with a Narrow Electrode Span"

_sensors, 2022, doi:10.3390/s22207743_

Round 1

Reviewer 1 Report

Attached file

Reviewer 2 Report

Dear Authors,

I have carefully reviewed your manuscript entitled “Identification of single yeast budding using impedance cytometry with a narrow electrode span” and it has impressed me. In my opinion, your work is innovative, relevant and sound. The manuscript is well written and organized and introduction and methodology let the reader have an idea of what you are researching and how do you proceed. Additionally, conclusions  are well supported by the results. So that, I can only congratulate you and suggest your manuscript for publication.

Nevertheless, please correct a couple of number-related typewriting errors in lines 258 (please write “show” instead of “shows”) and 285 (“differences” instead of “difference”)

Finally, I would like to encourage you to continue with your interesting research.

Best regards!!!

Reviewer 3 Report

Look the attachement

Round 2

Reviewer 1 Report

The revised manuscript is suitable for publication.

Author Response

we appreciate your valuable contribution
